# Impact of distinct FG nucleoporin repeats on Nup98 self-association

Alain Ibáñez de Opakua [1], Christian F. Pantoja [1], Maria-Sol Cima-Omori [1], Christian Dienemann [2] & Markus Zweckstetter [1,3] ✉

Nucleoporins rich in phenylalanine/glycine (FG) residues form the permeability barrier within the nuclear pore complex and are implicated in several pathological cellular processes, including oncogenic fusion condensates. The self-association of FG-repeat proteins and interactions between FG-repeats play a critical role in these activities by forming hydrogel-like structures. Here we show that mutation of specific FG repeats of Nup98 can strongly decrease the protein's self-association capabilities. We further present a cryo-electron microscopy structure of a Nup98 peptide fibril with higher stability per residue compared with previous Nup98 fibril structures. The high-resolution structure reveals zipper-like hydrophobic patches which contain a GLFG motif and are less compatible for binding to nuclear transport receptors. The identified distinct molecular properties of different regions of the nucleoporin may contribute to spatial variations in the self-association of FG-repeats, potentially influencing transport processes through the nuclear pore.

Phase separation has been reported as a driving force for the formation of cellular organelles not surrounded by membranes, such as nucleoli and stress granules[1–4]. Such organelles formed via phase separation assemble proteins into liquid-like "condensates" where protein concentration increases significantly[5]. Phase-separated condensates consist of multiple proteins and/or RNA. Many of the constituent proteins possess intrinsically disordered regions that facilitate phase separation through transient and multivalent interactions[6,7]. Intrinsically disordered regions, which often contain low-complexity domains, are typically composed of a limited set of amino acids. They lack a well-defined three-dimensional structure[8]. This often precludes insights into the molecular basis of their phase separation and associated activities.

Nuclear pore complexes (NPCs) are structures located within nuclear pores that serve as conduits connecting the nucleoplasm and cytoplasm. A single NPC facilitates the translocation of around 1000 molecules per second in both directions, enabling rapid material exchange[9]. Furthermore, NPCs play a crucial role as selective permeability barriers. Small molecules weighing ~30 kDa or less can passively diffuse through, while larger molecules are more difficult to pass and

generally require nuclear transport receptors for their passage[10]. NPCs are composed of large protein complexes and ~30 nucleoporins (NUPs)[11–13]. The nucleoporins situated in the central channel of NPCs are known as phenylalanine/glycine (FG)-NUPs due to their consecutive repeat of the FG motif[9,13–17]. Each NPC contains multiple copies of over 10 FG-NUPs, with numerous intrinsically disordered FG repeat domains filling the central channel of nuclear pores and forming the permeability barrier.

Different models have been suggested for the molecular organization of the NPC permeability barrier[18–21]. In some of these models, cohesive interactions between FG motifs do not play a role, while in others, cohesive FG–FG interactions are critical. In particular, in vitro reconstitution experiments showed that the intrinsically disordered FG repeat domain from yeast Nsp1 forms phase-separated hydrogel-like particles that reproduces the permeability barrier observed in NPCs[22,23]. Similar characteristics have been observed in other FG-NUPs[24,25]. The local concentration of FG motifs in the central plug of the nuclear pore in vivo is estimated to be 50 mM[22,26]. FG repeat domains may thus form gels even within NPCs of live cells. Electron microscopy of the hydrogel-like particles composed of FG repeat

[1]German Center for Neurodegenerative Diseases (DZNE), Von-Siebold-Str. 3a, Göttingen, Germany. [2]Max Planck Institute for Multidisciplinary Sciences, Department of Molecular Biology, Am Fassberg 11, Göttingen, Germany. [3]Max Planck Institute for Multidisciplinary Sciences, Department of NMR-based Structural Biology, Am Fassberg 11, Göttingen, Germany. ✉e-mail: Markus.Zweckstetter@dzne.de

domains from FG-NUPs revealed interlaced amyloid fibers that create a meshwork structure[27]. Additionally, cryo-electron microscopy (cryoEM) of amyloid fibrils formed by a 8-residue fragment of Nup98 (residues 116-123) provided molecular insights into structural interactions stabilizing amyloid fibrils of low-complexity domains[28]. Solid-state nuclear magnetic resonance (NMR) of nucleoporin FG gels further indicated that regions containing asparagine and glutamine residues form cross-β-structure[29]. Structural analyses of NPCs using high-speed atomic force microscopy and cryo-electron tomography also suggested that FG-NUPs extend filamentous protrusions into the central channel[30,31].

In this work, the structure of the NPC scaffold that encloses the central channel has been resolved with near atomic resolution[32–36]. However, much less is known about the structural dynamic properties of the NPC permeability barrier formed by FG-NUPs[37]. Here, we combine phase separation experiments and site-directed mutagenesis to gain insight into the contribution of specific FG-repeats of Nup98, an important FG-NUP of the NPC and key component of oncogenic fusion condensates[38–41], to its aggregation and hydrogel-like particle formation. We further reveal the molecular interactions that determine the self-association of a Nup98 GLFG motif at high resolution using cryoEM.

## Results

### Aggregation propensity varies over the Nup98 sequence

The N-terminal 384 residues of human Nup98 (named Nup98[FG]) comprise two prion-like domains connected by an approximately 50-residue-long linker (Fig. 1a). The prion-like domains are rich in aromatic (phenylalanine) and polar (serine, threonine, asparagine and glutamine) residues, while the charged residues of Nup98[FG] are localized predominantly in the region connecting the two prion-like domains, which also contains an RNA transport factor-binding motif (Gle2-binding sequence; GLEBS). Nup98[FG] has a high density of FG-repeats comprising in total 41 phenylalanine residues in 31 FG motifs. 14 of the FG motifs are part of GLFG motifs with eight of them being canonical (Fig. 1b).

To gain insight into the propensity of different parts of Nup98[FG] to self-associate, we divided the Nup98[FG] sequence into 18 overlapping peptides and studied their aggregation propensity (data from our previous publication[42]). On this basis, we calculated the per residue aggregation propensity (Fig. 1c). Through this analysis, we identified two regions with higher tendency to aggregate: one around residue 100, and a second one around residue 310 (Fig. 1c). They are dominated by the two aggregation-prone peptides that comprise the Nup98 residues 85-124 and 298-327, respectively. The first peptide aggregates into amyloid fibrils which previously were structurally resolved to high-resolution by cryoEM (Fig. 1d). This structure contains a large cavity formed by polar residues and most of the GLFG-like motifs fold into a β-turn/β-arch structure with phenylalanine and leucine pointing to the same side[42].

### Self-association of Nup98[FG] depends on specific motifs

To investigate whether the more aggregation-prone regions of Nup98[FG] are important for the ability of Nup98[FG] to form aggregated particles, we introduced phenylalanine-to-serine mutations into GLFG motifs present inside and outside of these two aggregation-prone regions in full-length Nup98[FG]. We selected GLFG motifs for the mutational analysis, because the [100]SLFS[103] motif (non-canonical GLFG motif) strongly stabilizes the structure of amyloid fibrils formed by residues 85-124 of Nup98 (Fig. 1e, f). We thus recombinantly prepared Nup98[FG] mutants with a single phenylalanine mutated to serine in the [100]SLFS[103] motif of the first aggregation-prone region or the [317]GLFG[320] motif of the second aggregation-prone region, as well as a double mutant with both phenylalanines mutated (F102S, F319S). Additionally, we prepared a Nup98[FG] mutant carrying the F228S mutation in between the two aggregation-prone regions.

The self-association properties of the Nup98[FG] mutants together with the wild-type protein were characterized using a combination of NMR spectroscopy, circular dichroism (CD) and dynamic light scattering (DLS). The three biophysical methods efficiently complement each other as each of the three methods is particularly sensitive for distinct structural properties. One-dimensional [1]H NMR spectroscopy enables accurate quantification of monomeric/low molecular weight species of proteins in a time- and temperature-dependent manner. DLS is able to detect soluble high-molecular weight species of Nup98[FG] that may not contribute to NMR signals. CD is highly sensitive to the presence of β-structure which is formed in amyloid fibrils. Additionally, we stained the particles formed by Nup98[FG] and its mutant proteins by the dye thioflavin-T (ThT) in order to probe the presence of amyloid-like β-structure in these particles.

Nup98[FG] is soluble and predominantly disordered at pH 3 with some residual ß-structure[42]. When compared to the Nup98[FG] wild-type protein, the amount of ß-structure estimated from circular dichroism spectra decreases by ~15% and ~19% for the single (F102S and F319S, respectively) and by 30% for the double (F102S/F319S) mutant Nup98[FG], but only by ~7% for the control mutant F228S (Fig. 2c and Supplementary Fig. 1). We showed previously by NMR spectroscopy that FG motifs, and especially GLFG motifs, transiently populate ß-structure in the monomeric state[42]. The three motifs, which were mutated, represent around 15% of secondary structure propensity, which corresponds to ~6% of the total ß-structure of the monomeric wild-type protein.

Upon raising the pH to 7, Nup98[FG] rapidly clusters into particles which are not detectable by NMR. While 67 % of the wild-type protein forms aggregates just after changing the pH to 7, in the F228S mutant 46% of the protein aggregates. However, in the Nup98[FG] proteins that contain either one (F102S, F319S) or two mutations (F102S/F319S) in the aggregation-prone regions only 28 and 23 % of the protein aggregated, respectively (Fig. 2a, b). DLS performed at pH 7 confirmed that the amount of monomeric protein increases from wild-type to F228S, and further to F102S, F319S and F102S/F319S Nup98[FG] (Fig. 2d). Additionally, the wild-type, F228S and F102S Nup98[FG] samples contain some larger particles even at pH 3, which were not detected for F319S or F102S/F319S Nup98[FG] (Supplementary Fig. 2).

We next characterized the ability of the four Nup98[FG] proteins to aggregate into particles, which had been shown to have hydrogel-like properties[22,25] (Fig. 3). For wild-type Nup98[FG], differential interference contrast microscopy revealed many particles, which, upon exposing the sample to the amyloid-specific dye ThT, display fluorescence (Fig. 3; top row). Previous fluorescence recovery after photobleaching experiments showed little fluorescence recovery, suggesting that the internal structure of the particles is solid-like[42]. ThT-positive particles and particle clusters were also observed for the F228S and F102S mutants of Nup98[FG] (Fig. 3; second row). The combined data demonstrate that destruction of different GLFG motifs by F-to-S mutation decreases the aggregation propensity of Nup98[FG].

### CryoEM structure of Nup98[FG](298-327) shows zipper-like hydrophobic patches

To understand the mechanistic basis of the impact of the F319S mutation on the self-association and aggregation of Nup98[FG], we aggregated a 30-residue peptide, which comprises residues 298-327 of Nup98[FG], into amyloid fibrils. The three-dimensional structure of the Nup98[FG](298-327) fibrils was subsequently determined using cryoEM. CryoEM is a versatile method to determine high-resolution structures of macromolecular assemblies including amyloid fibrils[43].

CryoEM micrographs of Nup98[FG](298-327) fibrils were of high quality and displayed well-defined and mostly separated amyloid fibrils (Fig. 4a). Because the fibrils displayed very small twist, we estimated the cross-over by merging multiple boxes after 2D classification (Fig. 4b, c). The analysis resulted in a twist of 1.20°. We subsequently

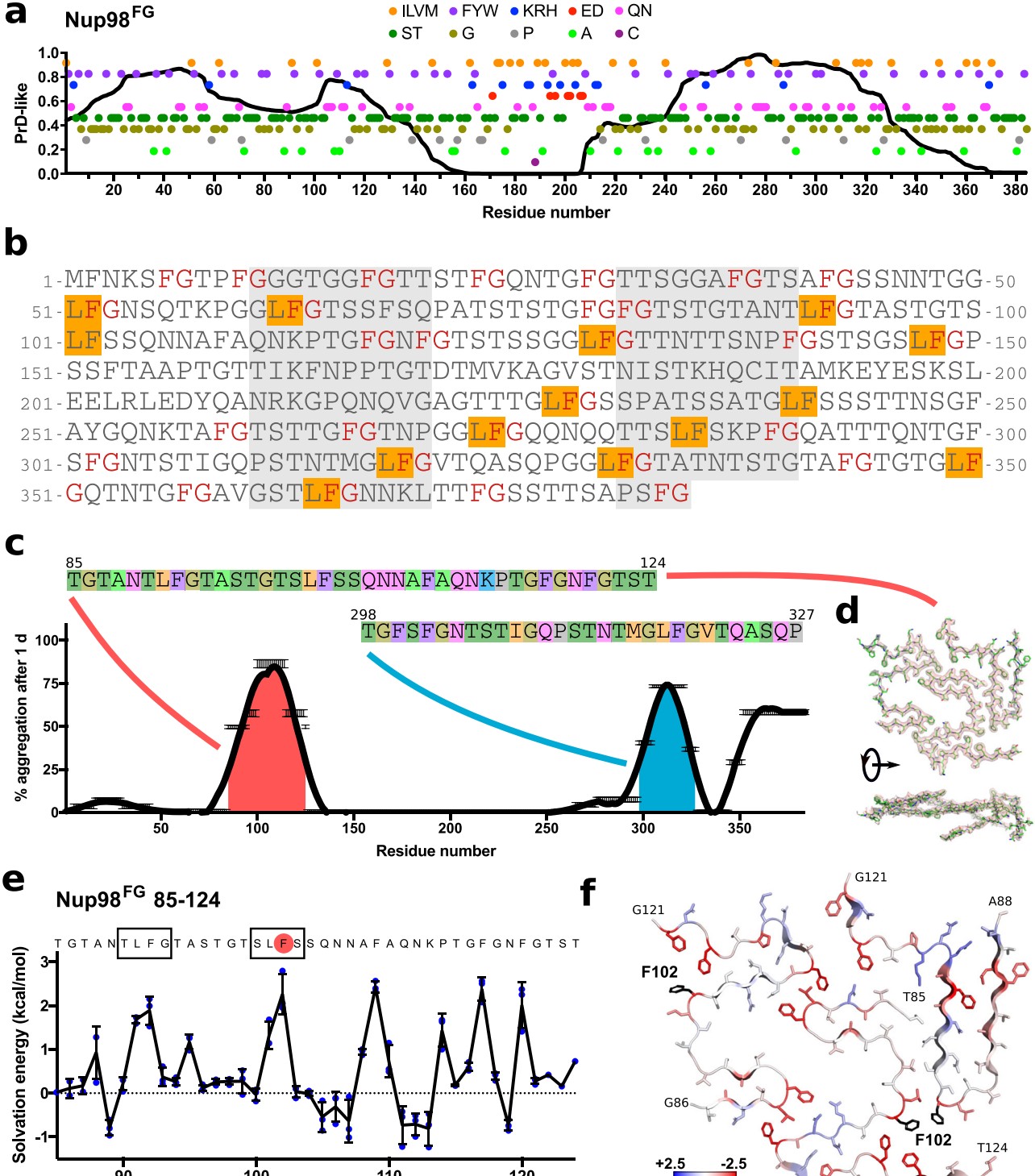

**Fig. 1 | Nup98 aggregation propensity depends on the primary sequence.**
**a** Representation of the prion-like domain (PrD-like) propensity[67] (black line) of Nup98 together with the distribution of specific residues over the sequence (hydrophobic, ILVM, orange; aromatic, FYW, purple; positively charged, KRH, blue; negatively charged, ED, red; polar with amide, QN, pink; polar with hydroxyl, ST, green; glycine, dark yellow; proline, gray; alanine, light green; cysteine, dark purple). **b** Amino acid sequence of the N-terminal part of Nup98 highlighting FG (red text) and LF (orange background) motifs. **c** Average aggregation per residue based on the aggregation of the analyzed peptides by NMR after one day at 5 °C with 2 mM peptide concentration. The sequences of the more aggregating peptides are represented with the residue colors from (**a**). **d** Cryo-EM structure (PDB-ID 7Q64) of the main polymorph formed by the first aggregation peptide Nup98FG(85-124). **e** Solvation energy over the sequence of the Nup98FG(85-124) peptide. LF motifs are represented into squares and the residue selected for mutation (F102) is highlighted in red. Each independent value from each of the three peptides is represented in blue. The error bars represent the standard deviation. **f** Structure of the aggregated Nup98FG(85-124) peptide with solvation energy in blue-red color code. Initial and last residues of each peptide are labeled. The mutated residue (F102) is highlighted in black. Source data are provided as a Source Data file.

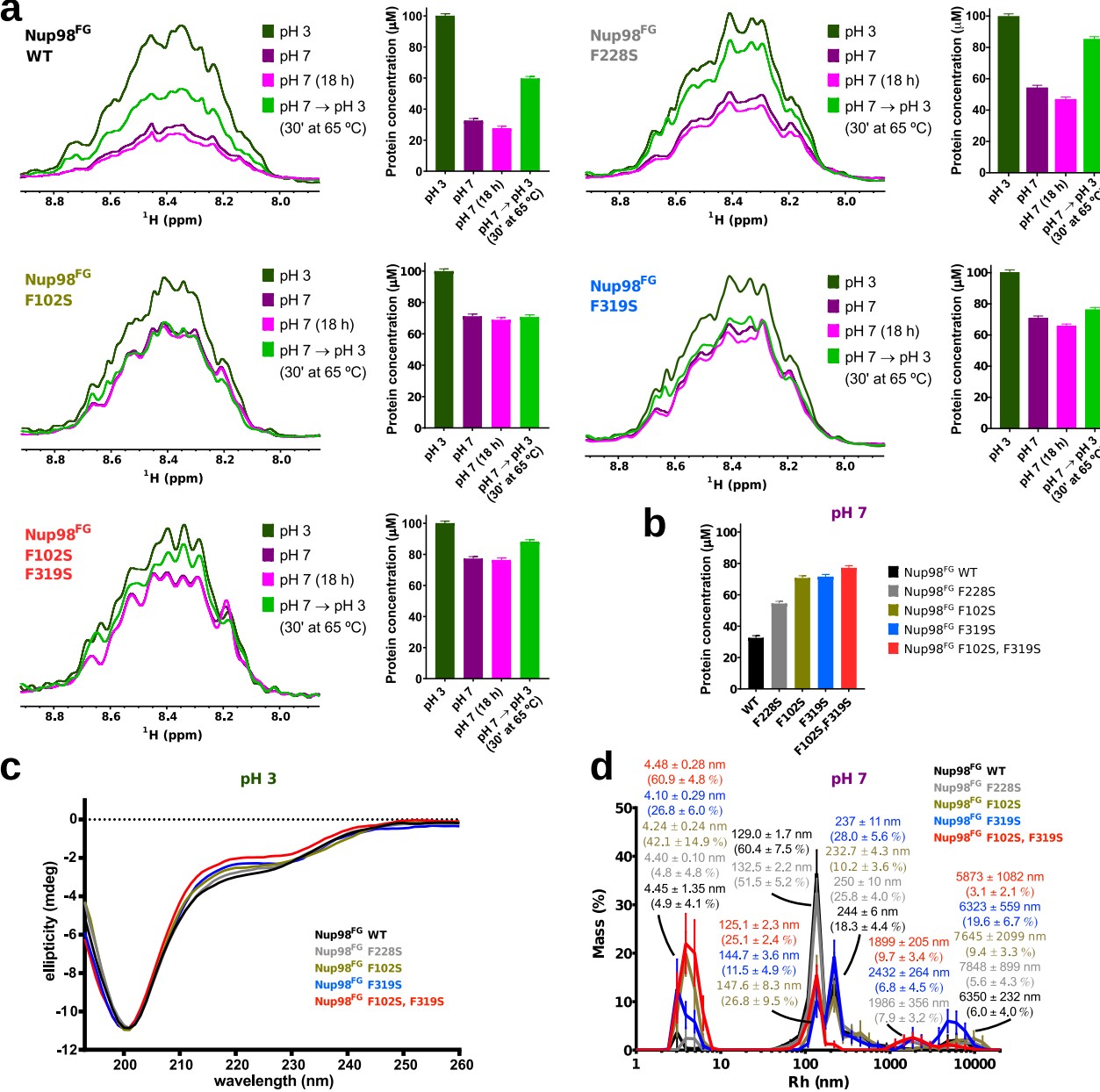

**Fig. 2 | FG-repeat mutations decrease β-structure and self-association of Nup98. a** Aggregation analysis of different Nup98[FG] constructs (WT, black; F228S, gray; F319S, blue and F102S + F319S, red) by NMR. For each mutant the ¹H-1D NMR spectra of the amide region at pH 3 (dark green), pH 7 (dark purple), pH 7 after 18 h (pink) and pH 3 after changing the pH back from 7 and incubating 30 minutes at 65 °C (green) is shown. The same color code is used to the right in a bar plot with the quantification of the soluble protein concentration in each condition. Error bars represent the error in the quantification based in the signal to noise ratio. **b** Soluble

protein concentration of each Nup98[FG] protein at pH 7. **c** CD spectra of the different Nup98[FG] constructs (WT, black; F228S, gray; F102S, brown; F319S, blue and F102S + F319S, red) at pH 3. **d** Hydrodynamic radii of Nup98[FG] mutants (WT, black; F228S, gray; F102S, brown; F319S, blue and F102S + F319S, red) at pH 7. Sizes, together with the population percentages in mass and the respective standard errors from 12 measurements, are displayed. Source data are provided as a Source Data file.

optimized the helical parameters and obtained a map at 2.67 Å resolution (Fig. 4d, e). Modeling and refinement resulted in a high-resolution structure of the amyloid conformation of Nup98[FG](298-327) (Fig. 4e and Supplementary Table 1).

Each layer of the amyloid fibril of Nup98[FG](298-327) is formed by five peptides in extended ß-structure-like conformation (Fig. 4e). Four of the molecules are oriented parallel and one, located at the edge, antiparallel. The three central molecules could be resolved from residue G299 to Q323/A324. Side-chains are tightly packed without the presence of internal cavities. The five molecules are located in a plane

giving rise to a flat sheet-like structure when viewed from the side of the fibril axis (Fig. 4e, lower part).

To gain insight into stability-contributing regions, we performed a solvation energy analysis of the fibril structure of Nup98[FG](298-327). The analysis points to two regions as being important for the stability of the fibril structure (Fig. 5a). One is generated by the ³⁰⁰FSFG³⁰³ motif, which forms a four-phenylalanine patch (Fig. 5b, purple). The second one comprises the ³¹⁷GLFG³²⁰ motif, which contains the mutated F319, and is flanked by two additional hydrophobic residues (methionine and valine). Together they form two zipper-like hydrophobic

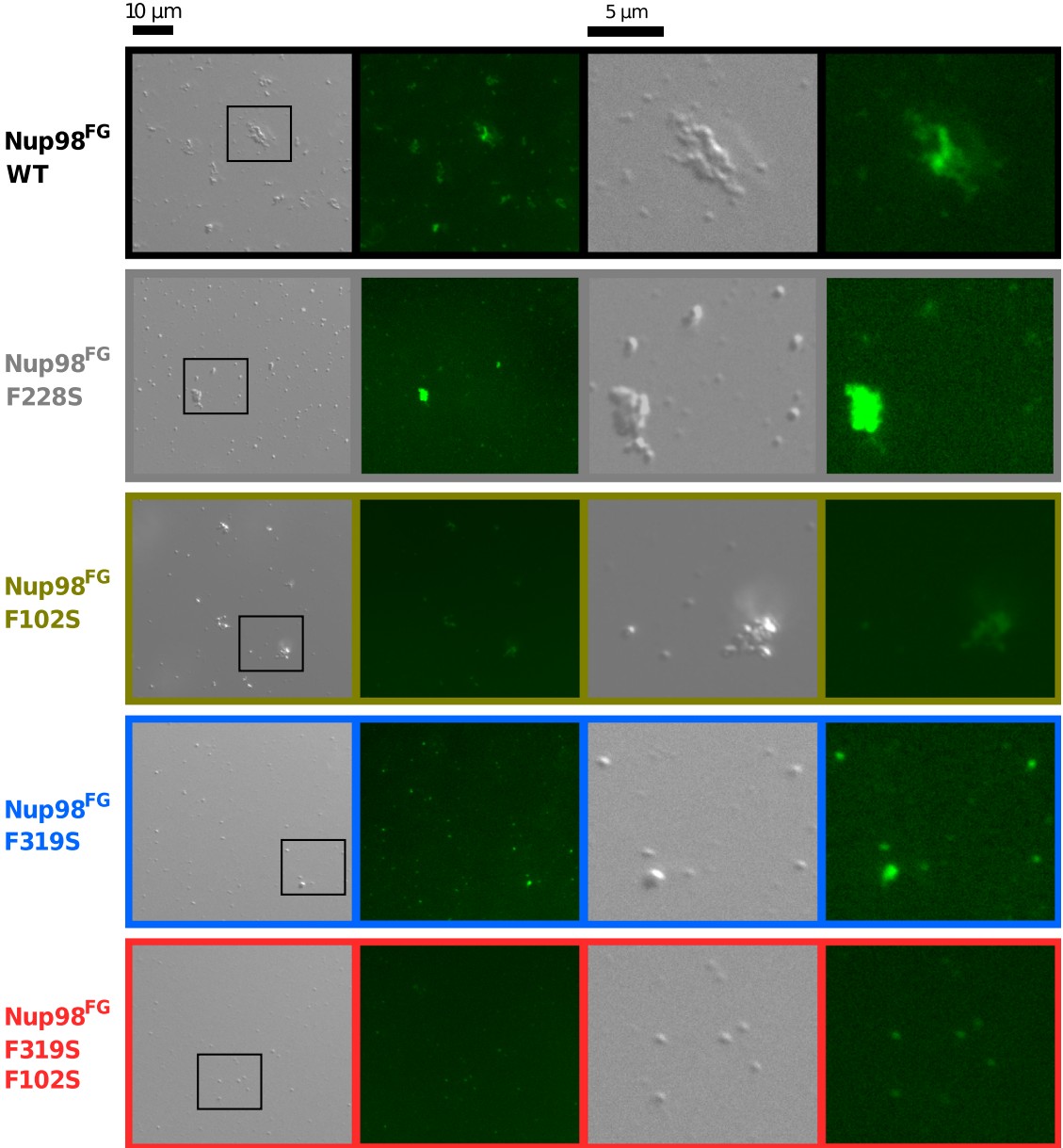

**Fig. 3 | Impact of Nup98 FG-repeat mutations on phase separation.** Differential interference contrast (left of each panel) and fluorescence (right of each panel) microscopy of different Nup98FG constructs (WT, black; F228S, gray; F102S, brown; F319S, blue and F102S + F319S, red) at pH 7 in presence of ThT. The right images display zoomed image regions. Experiments repeated at least twice with similar results and measuring WT as control at the same time.

arrangements in the structure of the Nup98FG(298-327) fibrils (Fig. 5b, orange). Residues in between these two motifs are mainly of polar nature (threonine, serine, asparagine, and glutamine), which can engage in hydrogen bonds and thus further stabilize the structure.

## Discussion

The human nucleoporin Nup98 plays an important role in multiple physiological and pathological processes[38–41,44–46]. Besides its role as key component of the NPC, the FG-repeat domain of Nup98 fuses with a chromatin-binding domain due to recurring chromosomal translocations in certain types of leukemia[40,41]. This fusion results in oncogenic properties, primarily linked to the formation of concentrated condensates[39]. The FG-repeat domain of Nup98 also facilitates the aggregation of the microtubule-associated protein tau, which is closely linked to Alzheimer's disease, resulting in the formation of tau-containing Nup98 aggregates in neurons[46]. Site-directed mutagenesis

studies revealed that the FG repeats of Nup98, and especially their phenylalanine residues, are essential for its self-association[47]. Detailed molecular insights into the self-association of the FG-repeat domain of Nup98 are thus critical to better understand its diverse physiological and disease-associated activities.

To understand the contribution of specific FG-repeats to nucleoporin self-association, we introduced F-to-S mutations in the two most aggregation-prone regions of the 384-residue long FG-domain of Nup98 (Fig. 1). In addition, we designed an F-to-S mutation in the intervening sequence. We found that mutating F228 to serine in this intervening region affected the self-association of Nup98FG to a smaller degree than the F102S or F319S single mutations present in the N-terminal and C-terminal part of Nup98FG, respectively (Figs. 2, 3). Consistent with this observation, the F228S mutant protein, as well as wild-type and F102S Nup98FG, undergo residual self-association even at low pH (Supplementary Fig. 2a). At neutral pH, where the self-

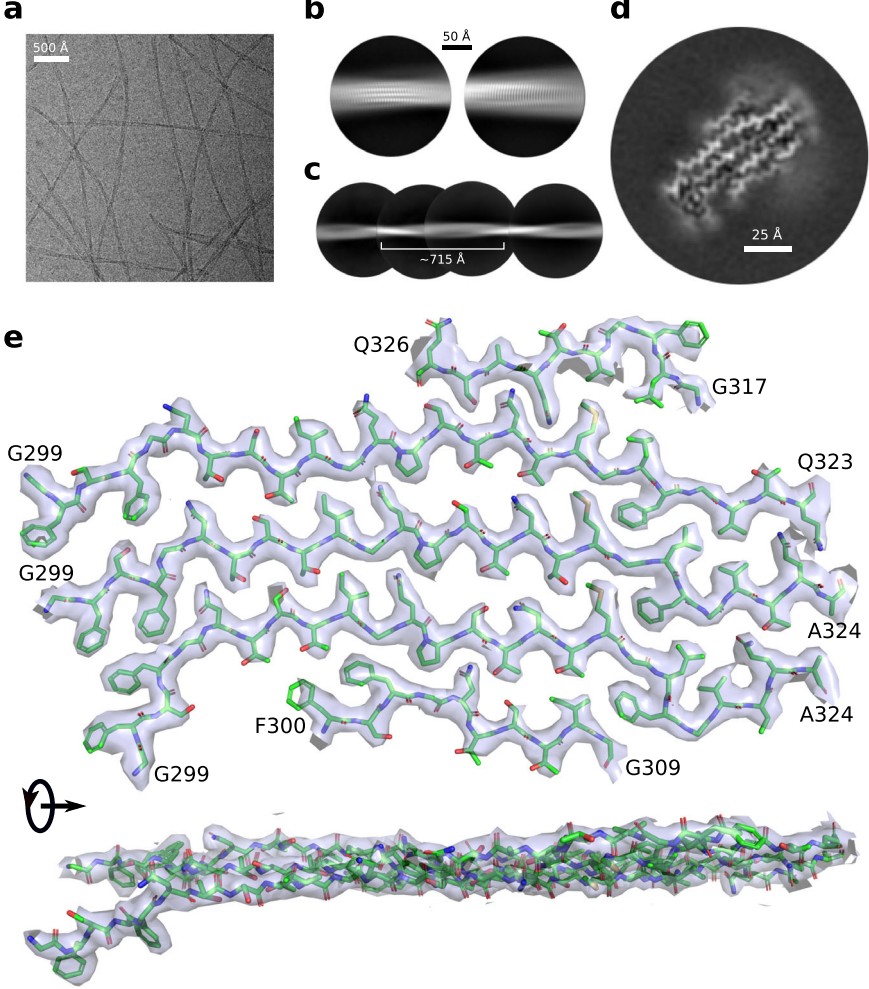

**Fig. 4 | Cryo-EM structure of Nup98^FG(298-327) fibrils. a** Cryo-electron micrograph of Nup98^FG(298-327) fibrils (1 of the total 11,290 micrographs). **b** Examples of high resolution 2D classes of the Nup98^FG(298-327) fibrils. Scale bar shown on top. **c** Reconstruction of a Nup98^FG(298-327) fibril from low-resolution and big box 2D classes for crossover estimation. **d** Cross-section of the cryo-EM map of the Nup98^FG(298-327) fibril after 3D refinement. Scale bar shown inside. **e** Single cross-section of the structure of Nup98^FG(298-327) fibrils together with the cryo-EM density map (from the top and the side of the fibril axis). Density maps were zoned for clarity, unzoned map is shown in Supplementary Fig. 3.

association of Nup98^FG is enhanced, the amount of aggregates that are formed by F102S and F319S mutant proteins was consistently lower than for the F228S-mutated Nup98^FG. For the double mutant the effect was even stronger but much less than the addition of both single mutations (Fig. 2b) suggesting a possible cooperative effect between both domains, i.e. both XLFX motifs are necessary together to acquire wild-type aggregate formation capabilities.

However, what are the molecular interactions that determine the ability of individual FG-repeats to self-associate? First insights into this question were obtained from a cryoEM structure of amyloid fibrils formed by an 8-residue fragment of Nup98 (residues 116-123)[28]. The structure revealed that the phenylalanine residue forms pi-stacks along the fibril axis and patches with phenylalanines of other peptides to stabilize the structure[28]. More recently we determined cryoEM structures of amyloid fibrils formed by a longer peptide that comprises the Nup98 residues 85-124, i.e. a peptide which overlaps with the N-terminal aggregation-prone region of Nup98^FG (Fig. 1)[42]. Nup98(85-124) folds primarily into short turn structures inside amyloid fibrils (Fig. 1d, f). The individual Nup98(85-124) molecules are further arranged such that a large cavity is present which is surrounded by polar residues. Consistent with the importance of phenylalanine residues for the self-association of nucleoporins, the phenylalanine residues of Nup98(85-124) form hydrophobic clusters together with nearby leucine residues in the amyloid structure (Fig. 1d, f).

The cryoEM structure of Nup98^FG(298-327) fibrils, however, reveals a very different molecular arrangement. These fibrils are formed by several aligned peptides folded into an extended conformation with no turns and without cavities (Fig. 4e). In addition, the side-chains of the Nup98^FG(298-327) residues are present in a zipper-like arrangement (Fig. 4e). The main stabilizing region comprises the ^317GLFG^320 motif flanked by two hydrophobic residues (MGLFGV). Because of the structural flexibility of glycine residues, the hydrophobic residues can point to either side of the chain to favor the formation of hydrophobic patches. For example, in the top hydrophobic patch of Fig. 5b, the valine of the top chain is oriented in the opposite direction of the valine of the bottom chain and this enables their interaction. The other hydrophobic residue of the ^316MGLFGV^321 patch is methionine, which is known to participate in the formation of labile cross-β polymers in a redox-dependent manner[48].

The large structural differences, which we observe in the amyloid-conformations of the two different FG-rich regions of Nup98, have a small effect on the aggregated particle formation behavior of Nup98^FG. The F102S mutant presents a smaller decrease in ß-structure than F319S (also detectable by the appearance of larger particles by DLS)

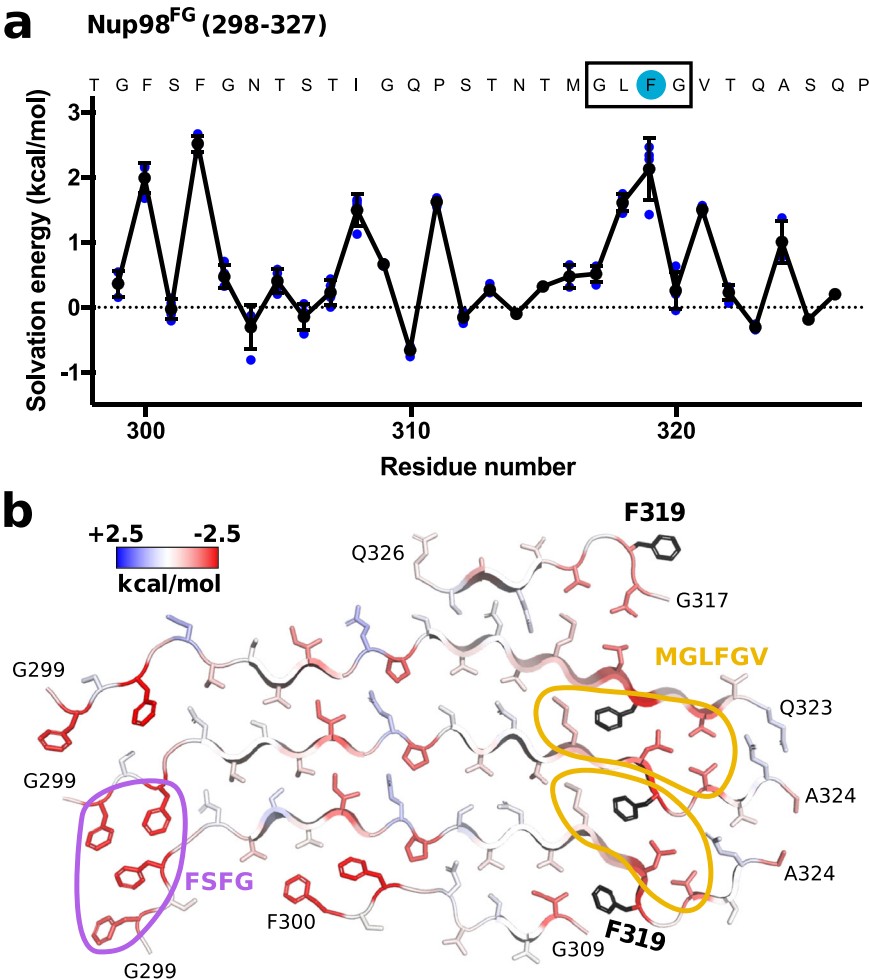

**Fig. 5 | Nup98^{FG}(298-327) fibril structure is stabilized by hydrophobic patches.** **a** Residue-specific solvation energies in the Nup98^{FG}(298-327) fibril structure. The LF motif is boxed; the mutated residue (F319) is highlighted in blue. Each independent value from each of the 5 peptides is represented in blue. The error bars represent the standard deviation. **b** Structure of the Nup98^{FG}(298-327) fibril with solvation energy in blue-red color code. F319, which was mutated, is highlighted in black. Main hydrophobic patches are highlighted in the structure (purple for FSFG sequence, and orange for the MGLFGV sequence). Source data are provided as a Source Data file.

and keeps slightly less monomeric protein in our NMR experiments. This change goes in the direction of the fact that the Nup98(85-124) fibrils have less packed side-chain conformations and display lower stability per residue (−0.51 kcal/mol) when compared to Nup98^{FG}(298-327) fibrils (−0.65 kcal/mol). On the other hand, the conformations of the GLFG motifs, which are present in the Nup98(85-124) fibrils are highly similar to the conformation of GLFG motifs in complex with nuclear transport receptors[42]. These structural properties may facilitate the binding of nuclear transport receptors to these FG-repeats in the central NPC channel[42,49]. In contrast, the tight zipper-like arrangement of the FG-repeats in the Nup98^{FG}(298-327) fibrils facilitates hydrophobic patch formation with higher stability per residue (Fig. 5). This may make it more difficult for nuclear transport receptors to open the FG-repeat meshwork in order to transport cargo across the NPC permeability barrier. Alternatively or complimentary, more liquid-like states of Nup98^{FG} repeats – potentially regulated by post-translational modifications such as O-GlcNAcylation[50]- may impart specific structural properties inside the NPC channel[19]. Consequently, distinct FG-repeat regions may generate spatially distinct zones that facilitate at different levels the access of nuclear transport receptors, potentially/possibly influencing molecule transport processes through the NPC[18,51–53]. However, further in vitro and in vivo experiments will be required to confirm these hypotheses and especially to establish the physiological role of these Nup98 fibrillar structures.

## Methods

### Protein preparation

The FG-repeat domain of human Nup98 (Nup98^{FG}; residues 1-384) was cloned into a bacterial expression vector pET28a (Novagen) between the restriction sites NheI and XhoI, keeping the N-terminal HisTag for its purification.

Nup98^{FG} mutants (F102S, F228S, F319S, F102S/F319S) were obtained by using the QuikChange Site-Directed Mutagenesis method (Stratagene). Polymerase chain reactions (PCR) were performed with Phusion High-Fidelity Polymerase (ThermoFisher) and the resultant DNA product was treated with DpnI (NEB) to digest the parental methylated DNA, and transformed into NEB 5-alpha competent *Escherichia coli*. A subsequent PCR using the verified single mutant F319S was run to obtain the double mutant F102S/F319S. Introduced mutations were verified by DNA sequencing. Primers for the site-directed mutagenesis reported in Supplementary Table 2.

Recombinant Nup98^{FG} proteins were expressed in the *Escherichia coli* expression strain BL21 (DE3). Bacteria were grown in LB medium supplemented with kanamycin to an $OD_{600}$ of ~0.8, induced with 0.5 mM IPTG and incubated overnight at 37 °C. The harvested cell pellets were sonicated in lysis buffer containing 100 mM $Na_2HPO_4$, 10 mM Tris and 10 mM 2-mercaptoethanol at pH 8.5; the lysate was clarified by centrifugation. After centrifugation, the resultant pellet containing Nup98^{FG} was resuspended in denaturing buffer containing

8 M Urea, 100 mM $Na_2HPO_4$, 10 mM Tris and 10 mM 2-mercaptoethanol at pH 8.5. After a second centrifugation step, the supernatant was loaded onto a self-packed Ni-NTA column (Qiagen) equilibrated with denaturing buffer and the bound protein was eluted with 6 M Urea, 100 mM $Na_2HPO_4$, 10 mM Tris and 10 mM 2-mercaptoethanol at pH 4.0. The same elution buffer was used to run a size exclusion chromatography, Superdex75 26/600 (GE Healthcare). A second size exclusion chromatography was performed to remove the denaturation conditions and to improve the purity of the sample by using 50 mM sodium phosphate buffer pH 3.0 and 1 mM TCEP. The pure proteins were concentrated by ultracentrifugation with a 5 kDa MWCO membrane.

Nup98 peptides were prepared by solid-phase synthesis (Genscript). For cryo-EM measurements, powder of lyophilized Nup98$^{FG}$(298-327) peptide was dissolved in pure water to reach a concentration of 2 mM, followed by incubation at 25 °C for one day.

### Circular dichroism
Nup98$^{FG}$ samples were prepared at 5 µM in sodium phosphate buffer 50 mM, TCEP 1 mM, pH 3 and incubated 30 minutes at 65 °C before measurement. The pH 7 experiments were measured with the same samples after adding NaOH to adjust the pH to 7. CD data were collected from 185 to 280 nm using a Chirascan-plus qCD spectrometer (Applied Photophysics, Randalls Rd, Leatherhead, UK) at 25 °C, 0.5 time-per-point (s) in 1 nm steps. The datasets were averaged from 10 repeats. Spectra were baseline corrected against buffer and smoothened (window size: 4).

### Dynamic light scattering
Nup98$^{FG}$ wild-type and mutant protein samples for DLS were prepared at 2.5 µM in sodium phosphate buffer 50 mM, TCEP 1 mM, pH 3 and incubated 30 minutes at 65 °C before measurement. The pH 7 experiments were measured with the same samples after adding NaOH to adjust the pH to 7 and subsequently decreasing the protein concentration, if needed, to avoid saturation of the detector (to 100 nM for WT and F228S and 1 µM for F102S and F319S). Measurements were performed at 25 °C using a DynaPro NanoStar instrument (Wyatt Technologies Corporation) using NanoStar disposable MicroCuvettes. The samples were illuminated with a 120 mW air launched laser of 662 nm wavelength and the intensity of 90° angle scattered light was detected by an actively quenched, solid-state Single Photon Counting Module (SPCM). Data were acquired with an acquisition time of 5 s and a total of 5 acquisitions per measurement. Hydrodynamic radii were determined using the software package Dynamics 7.10.0.23. The final values are shown as the average and standard error of 12 measurements.

### Light microscopy
Nup98$^{FG}$ wild-type and mutant protein samples were prepared at 50 µM in sodium phosphate buffer 50 mM, TCEP 1 mM, pH 3 and incubated 30 minutes at 65 °C before measurement. The experiments were measured immediately after adding NaOH to adjust the pH to 7. ThT was added to reach a concentration of 50 µM. A total of 5 µl of the sample was loaded onto a slide and covered with a 18 mm coverslip. Differential interference contrast images as well as fluorescent images were acquired on a Leica DM6B microscope with a 63x objective (water immersion) and processed using ImageJ.

### NMR spectroscopy
NMR spectra were recorded at 5 °C on Bruker 900 MHz spectrometer equipped with triple-resonance cryogenic probe. The 1D NMR experiments were collected using a WATERGATE scheme under the Bruker pulse program zggpw5[54]. They were recorded with an acquisition time of 0.6062 s, spectral width of 15.01 ppm, and a relaxation delay of 1 s. Nup98$^{FG}$ wild-type and mutant protein samples were

prepared at 100 µM in sodium phosphate buffer 50 mM, TCEP 1 mM, pH 3 and incubated 30 minutes at 65 °C before measurement. The pH 7 experiments were measured immediately after adding NaOH to adjust the pH to 7 (deadtime of ~5 min). Then, after 18 hours, the sample was measured again, the pH changed back to 3 with HCl and the sample incubated 30 minutes at 65 °C in a water bath before the last measurement. For each sample one-dimensional (1D) $^1H$ spectra were measured in each condition, processed and the amide region integrated with TopSpin 3.6.1 (Bruker) to calculate the amount of soluble monomer in the sample. Errors were calculated from signal to noise.

For the estimation of aggregation propensities (Fig. 1c), we obtained the percentage of aggregation after one day for each peptide in the NMR measurement conditions. The aggregation propensity of each residue is determined by averaging the aggregation propensities of the peptides containing that residue.

### Cryo-electron microscopy
Nup98$^{FG}$(298-327) fibrils were prepared by dissolving the peptide in water to reach a concentration of 2 mM, followed by incubation at 25 °C for one day. Subsequently, sample volumes of 3 µl were applied to freshly glow-discharged R2/1 holey carbon grids (Quantifoil) and vitrified in liqid ethane using a Mark IV Vitrobot (Thermo Fischer Scientific) operated at 100% rH and 4 °C. Cryo-electron microscopy was done with a Titan Krios transmission-electron microscope (Thermo Fisher) operated at 300 keV accelerating voltage. Images were recorded at a nominal magnification of 81,000 x using a Quantum LS energy filter (Gatan) with slit with set to 20 eV and a K3 direct electron detector (Gatan) in non-super-resolution counting mode, corresponding to a calibrated pixel size of 1.05 Å on the specimen level. In total, 11,290 images with defocus values in the range of −0.7 µm to −2.0 µm were acquired in movie mode with 2.3 s acquisition time. Each movie contained 40 frames with an accumulated dose of approximately 40.62 electrons per Å². The resulting dose-fractionated image stacks, containing all frames 1-40, were subjected to beam-induced motion correction and CTF-estimation on-the-fly using Warp[55].

Manual fibril picking was done with EMAN2[56] e2helixboxer to select an average of around 20 segments per micrograph in 40 micrographs. The manual picking was used to train a model and pick the rest of the micrographs with crYOLO[57] with an inter-box distance of 19 Å. For fibril picking, only micrographs with an estimated resolution ≤4.0 Å and an average frame-to-frame motion across the first 16 frames <3.5 Å (10,304 micrographs) were considered.

Nup98$^{FG}$(298-327) fibrils were reconstructed using RELION-3.1.2[58], following the helical reconstruction scheme[59]. For an initial 2D classification, we extracted particle segments using a box size of 576 pixels downscaled to 96 pixels. Best classes with similar twist were selected and used to estimate a crossover of around 715 Å (Fig. 4c), which is equivalent to a twist of around −1.2° for a 4.75 Å rise. For 3D classification, the segments after 2D classification were re-extracted without downscaling using a box size of 224 pixels. We performed several rounds of 3D classification starting from a 60 Å low-pass-filtered featureless cylinder and subsequent 3D refinements to optimize the helical parameters (rise of 4.71 Å and twist of −1.19°, reported in Supplementary Table 1). Next, standard RELION post-processing with a soft-edged solvent mask that includes the central 10 % of the box height yielded the final post-processed map (sharpening B-factor of −80.66 Å²). The resolution (2.67 Å) was estimated from the value of the FSC curve for two independently refined half-maps at 0.143[60,61] (Supplementary Fig. 4).

The atomic model of Nup98$^{FG}$(298-327) fibrils were built de novo in Coot[62]. The high resolution of the cryo-EM maps allowed reliable modeling the protein backbone and side-chain rotamers. Refinement in real space was conducted using PHENIX[63,64] and Coot[62] in an iterative manner. The resulting models were validated with MolProbity[65]. Details of the atomic models are described in Supplementary Table 1.

## Solvation energy calculation

Stability calculations based on solvation energy were performed with the software accessiblesurfacearea_v07.2d[66]. Five layers were used for the Nup98$^{FG}$ 298-327 and nine for the Nup98$^{FG}$ 85-124 because of the less planar structure of its layers. Energy values from the middle layer were used.

## Reporting summary

Further information on research design is available in the Nature Portfolio Reporting Summary linked to this article.

## Data availability

The cryo-EM map generated in this study has been deposited in the Electron Microscopy Data bank (EMDB) under accession code EMD-16671 (Nup98(298-327) fibril). The corresponding atomic model has been deposited in the Protein Data Bank (PDB) under accession code 8CI8 (Nup98(298-327) fibril). The previous structure is available in the Protein Data Bank (PDB) under accession code 7Q64 (Nup98(85-124) fibril polymorph 1). Source data are provided as a Source Data file. Source data are provided with this paper.

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

## Acknowledgements

We thank David S. Eisenberg and Michael Sawaya for the code to calculate atomic solvation energies. Access to the FEI Titan Krios cryo-electron microscope with K3 detector was gratefully provided by Patrick Cramer at the MPI for Multidisciplinary Sciences, Goettingen, Germany. M.Z. was supported by the European Research Council (ERC) under the EU Horizon 2020 research and innovation program (grant agreement No. 787679).

## Author contributions

A.IdeO. performed biochemical experiments, NMR spectroscopy, fluorescence microscopy, prepared samples for cryo-EM, determined the cryo-EM structure and analyzed fibril structures. C.F.P. performed biochemical experiments, NMR spectroscopy, and fluorescence microscopy experiments. C.D. recorded the cryo-EM data sets. M-S.C-O. prepared recombinant Nup98^FG constructs. A.IdeO. and M.Z. designed the project.

## Funding

## Competing interests

The authors declare no competing interests.
