## [Peer Review File · Nature Communications]

Impact of distinct FG nucleoporin repeats on Nup98 self-associationREVIEWER COMMENTS

Reviewer #1 (Remarks to the Author):

Impact of distinct FG nucleoporin repeats on Nup98 self-association

Alain Ibanez de Opakua, Maria-Sol Cima-Omori, Christian Dienemann, and Markus Zweckstetter

Summary Paragraph: Intrinsically disordered regions (IDRs) are commonly found in proteins that phase separate and are frequently associated with diseases like cancer and neurodegeneration. Because disordered regions lack a predictable three-dimensional structure their precise function in phase separation is difficult to determine. The intrinsically disordered FG-repeat domain of FG-NUP proteins is thought to contribute to the size exclusion barrier formed by the phase separated nuclear pore compartment. The FG repeat of NUP98 also forms fusion oncoproteins whose role in cancers is thought to be driven by its propensity to phase separate. The current study focuses on the role of specific FG-residues in the aggregation of NUP98. Using a combination of site-directed mutagenesis and NMR the authors measure the contribution of distinct residues to the self-association of the protein. Cryo-EM of two different NUP98 peptide fibrils suggests a potential structural basis of nuclear pore transport.

Assessment: This study makes use of state-of-the-art structural biology techniques and applies them to a topic of high interest- how IDRs contribute to phase separation and aggregation. The experiments are of high quality, but some additional quantification is needed. Especially considering NUP98 fusion oncoproteins, those in the fields of cancer, phase separation, and general cell biology will be interested in the current study. With this broad audience in mind, parts of the manuscript should be reorganized for clarity and more experimental detail is needed in the text. I make the following specific suggestions for improvement:

1. The rationale for analyzing the F residues chosen is unclear when the solvation energy of other F residues within Nup98FG(85-124) (Figure 1E) and Nup98FG(298-327) (Figure 5A) was approximately equal to or greater than that of F102 and F319.
2. Why was the F102S mutation not examined independent of the F319S mutation? This omission makes it difficult to compare the two aggregation-prone regions of Nup98FG.
3. Quantification of this experiment should be included. Is ThT fluorescence indicative of phase separation or aggregation or both in this context?
4. The introduction was very clear and well-written. Suggestions to help with clarity of the results: consider re-organizing the paper to present the cryo-EM structure of the two peptide fibrils at the beginning. Propose the important residues based on the structure/solvation energy (Figure 1, 4, 5), and then introduce the mutations and show their effect on aggregation (Figures 2 and 3). The results section should also include more experimental detail and explanation to cater to a wider, non-structural biology

audience. For example, include one sentence to highlight the utility of each method for the study of IDRs (especially methods used in Figure 2 and the description of the cryo-EM in Figure 4) prior to stating the result. A final schematic that models how the distinct properties of Nup98(85-124) and Nup98(298-327) might contribute to the nuclear pore meshwork/aggregation would be helpful.

Minor:

Figure 1:

1. Given that other types of IDRs contain unique motifs (Ex:RGG repeats), directly comparing the contribution of the FG vs GLFG motif within Nup98FG(298-327) in the aggregation analysis would be of interest.
2. The cryo-EM structure in D is very small and is not discussed in detail like the one in Figure 4E.

Figure 2:

1. Consider including a graph that combines all results at a given pH to make it easier for the reader to directly compare the aggregation of WT and mutants.
2. In b there are clearly differences in ellipticity between WT and mutants, but the significance of these differences is unclear.

Figure 3:

1. Do the Nup98FG mutants aggregate less or just on a much slower time scale?

Figure 5:

1. It is unclear why the solvation energy of Nup98FG(298-327) isn't presented until Figure 5 when I believe it's used as rationale for choosing the F319S mutation. Consider adding it to Figure 1 along with the Nup98FG(85-124) data and using the last figure for a model schematic.

Reviewer #2 (Remarks to the Author):

Review: Impact of distinct FG nucleoporin repeats on Nup98 self-association

Corresponding author: Markus Zweckstetter (MPI Göttingen)

In this study, Ibáñez de Opakua et al. use nuclear magnetic resonance (NMR) and cryo-electron microscopy (cryoEM) based techniques to probe the structure and biophysical properties of a phenylalanine-glycine (FG) repeat region of the nucleoporin Nup98. The authors determined a 2.67Å cryoEM structure of an amyloid fibril obtained from a Nup98 fragment (residues 298-327) and analyzed the effect of mutations on Nup98's propensity to aggregate by collecting 1H-NMR spectra at different pH values as a proxy for aggregation, circular dichroism, dynamic light scattering, and by staining with the amyloid-binding dye thioflavin-T.

The main function of FG repeats in the nuclear pore complex (NPC) is to establish a permeability barrier

for the selective transport of macromolecules between the nucleus and the cytoplasm. Intrinsically disordered FG repeat polypeptides line the NPC's transport channel and are thought to form a mesh-like phase that blocks or hinders the free diffusion of proteins and ribonucleic acids in a size-selective manner. In-vitro reconstituted FG repeats have been shown to form liquid-liquid phase separated condensates (Celetti et al., JCB, 2020), which can mature into hydrogels (for example, Frey and Görlich, Cell, 2009 or Schmidt and Görlich, eLife, 2015), and amyloids (Halfmann et al. Prion, 2012; and Alberti et al. Cell, 2009). NMR techniques have been previously used to characterize cohesive FG interactions (Najbauer et al., Nat. Commun., 2022; Labokha et al., EMBO J., 2013; and Ader et al., PNAS, 2010). The current study is a direct extension of the authors' previous work. Previously, Ibáñez de Opakua et al. generated 18 overlapping fragments of a FG repeat region of Nup98 (residues 1-384) which they analyzed by NMR over time (Nature Chemistry, 2022). The rate of NMR signal loss allowed them to score different regions of Nup98 for their tendency to aggregate, and they came up with two "aggregation prone" regions: residues 85-124 and 298-327. In this 2022 study, the authors reported a biophysical analysis of the Nup98 85-124 fragment using NMR, circular dichroism, and physical calculations, and a high-resolution structure of the amyloid fiber obtained by cryoEM. Now, they repeated a similar analysis of the Nup98 298-327 fragment.

The manuscript contains many conclusions that are speculative and are not supported by experimental evidence. They should be thus removed. Some examples include:

Pages 7-10 (Results and Discussion): The interpretation of the impact of mutations introduced by the authors on Nup98 phase separation. The authors generated the following point mutations: F319S in the "second" aggregation-prone region, F102S/F319S that combines F319S with the F102S mutation in the "first" aggregation-prone region, and F228S in between the two aggregation-prone regions. They found that the Nup98 1-384 fragment is less prone to aggregation if harboring the F102S/F319S mutation. To substantiate the claims that "The combined data demonstrate that destruction of different GLFG motifs by F-to-S mutation decreases the aggregation propensity of Nup98FG. However, a particularly strong effect is exerted by the F319S mutation..." and that "The data show that individual FG-repeats have variable importance for the self-association capabilities of Nup98" the authors would need to analyze the F102S mutant in side-by-side experiments, as well as combinations of F319S with mutations they expect to have a milder effect, such as F228S/F319S. The claims are speculative given the current design of experiments.

Pages 11-12 (Discussion): The authors speculate that the two aggregation-prone regions that they have identified (residues 85-124 and 298-327) assume distinct roles in hydrogel formation and interactions with transport factors based on the packing of hydrophobic residues in the amyloid fibril cryoEM structures they determined. These claims are not substantiated by any experimental evidence, such as systematic mutagenesis of these regions followed by assays for hydrogel formation and transport factor binding. More fundamentally, there is no guarantee that the structures determined by the authors inform us on the interactions that give rise to Nup98 hydrogels in the cell. Work by Scheres et al. and by the authors themselves on the Nup98 85-124 fragment has shown that the same polypeptides aggregate into a variety of amyloid fiber structures depending on the aggregation conditions.

Page 10 (Discussion): There is no clear link between the presence of Nup98 in tau aggregates associated with Alzheimer's disease or the oncogenic Nup98 fusions and the aggregates or amyloid fibrils studied by the authors, so it is not clear how this study advances the understanding of Nup98's role in these diseases.

A significant issue with the manuscript is that the referencing of previous work is incomplete and overall arbitrary. Some examples include:

Page 2 (Introduction): In general, because the authors studied the human Nup98, all citations discussed in the introduction to the NPC should reference the human NPC instead of the *S. cerevisiae* NPC. For example, Kim et al. (Nature, 2018) is not appropriate for the claim “Each NPC contains multiple copies of over 10 FG-NUPs...”.

Page 2 (Introduction): Lin et al. (Science, 2016) should be cited in relation to “The nucleoporins situated in the central channel of NPCs are known as phenylalanine/glycine (FG)-NUPs due to their consecutive repeat of the FG motif.”

Page 2 (Introduction): The claim that “This hydrogel allows low molecular weight proteins and nuclear transport receptors to pass while impeding the passage of proteins weighing above 30 kDa.” is too simplistic for a manuscript about FG properties. The authors should consult Ribbeck et al. (EMBO J, 2001)

Page 3 (Introduction): The reference #27 (Milles et al., EMBO Rep., 2013) is not applicable to the claim “Solid-state NMR analysis of nucleoporin FG gels further...”. Is this a typo?

Page 3 (Introduction): The references associated with the claim that “structural analyses of NPCs using high-speed atomic force microscopy and cryo-electron tomography demonstrated that FG-NUPs extend filamentous protrusions into the central channel” do not actually “demonstrate” that what had been observed is an FG nucleoporin per se.

The authors need to discuss, in the introduction and discussion sections, the previously determined amyloid structure of a Nup98 fragment (residues 116-123) by Hughes et al. (2018, Science) and how the interactions captured by this fragment relates to the structures obtained from the Nup98 85-124 and 298-327 fragments.

Lastly, the physiological relevance of the Nup98 FG repeat structures determined by Ibáñez de Opakua et al. in the current and their previous manuscript (Nature Chemistry, 2022) needs to be established. Under what conditions do these particular amyloid fibers occur in the human body? Transport factors and chaperones have been shown to prevent the formation of FG repeat aggregates (Springhower, Rosen, and Chook, 2020, Curr. Opin. Cell Biol.; Prophet et al., 2022, Nat. Cell Biol.; and Kuiper et al., 2022, Nat. Cell Biol.). The structural variability of different Nup98 FG-repeat region fragments, as well as variability of structures that arise from the same 85-124 fragment, raises the question of what the actual structure of the intact FG repeat region is and what structures can occur in vivo. The lack of physiological validation substantially diminishes the excitement stemming from these structures, as it is unclear what can be learned about NPC (dys)function.

The currently presented data, which includes a structure of an amyloid fiber obtained from a Nup98 FG-repeat fragment (residues 298-327) and a mutational analysis that lacks rigor, is a modest advance with little conceptual novelty over the previous data, which included a mapping of aggregation propensity for the entire FG repeat region as well as structures and a mutational analysis of a different Nup98 FG-repeat fragment (residues 85-124). Many conclusions represent speculations that are not supported by experimental evidence. Unfortunately, the results only provide an incremental advance from previously published work, making this paper more appropriate for a more specialized journal.

Reviewer #3 (Remarks to the Author):

Ibanez de Opakua et al. present a manuscript in which they analyze the aggregation of the FG-nucleoporin Nup98. Based on aggregation assays using small Nup98 peptides, the authors calculate a per-residue aggregation propensity and thereby identify two regions within the protein that they predict to be particularly prone to aggregate. Through the insertion of specific Phe to Ser mutations into the aggregation prone regions (and the region in between), the authors aim to address whether GLFG motifs, important for hydrogel formation and nucleocytoplasmic transport, are also important for self association. Several aggregation assays and assays addressing secondary structure content (1D 1H NMR, CD spectroscopy, DLS, ThT staining and imaging of aggregates) are presented and confirm that Phe to Ser mutations decrease the aggregation propensity. Nup98 peptides comprising the two aggregation prone regions, respectively, were shown to form fibers and the structure of those was analyzed using electron microscopy. A detailed structural analysis of the fibers formed by the second aggregation prone peptide is presented and reveals the participation of the GLFG motif. The authors discuss the structural differences between the fibers formed by the two aggregating peptides and their potential impact for accessibility of FG-repeats to nuclear transport receptors and in the context of oncogenic fusion proteins.

The authors present a solid piece of work studying the aggregation of Nup98 by different methods and providing molecular insights into its aggregation behavior, which is of potential relevance to nucleocytoplasmic transport and oncogenic fusion condensates.

However, a few points need to be addressed:

- Based on an aggregation assay developed in Ibanez de Opakua et al. (2022), the authors calculate a per residue aggregation propensity. As the peptides used for this assay seem to be of different length and only partially overlapping, it is not clear how this has been done. Can the authors report how the per residue aggregation propensity was calculated?
- Two aggregation prone regions are identified, of which one is very similar to the one already studied in Ibanez de Opakua et al. (2022). The authors identify the structure of the amyloid fibers formed by that peptide, but do not report on any of their features or further analysis. If the authors decide to report on the newly determined structure of Nup98 85-124, they should dedicate more than one sentence to its analysis.
- This is particularly true, when the structural implications of the mutations are discussed, which is focused on the second peptide and F319S. No particular attention is given to F102S. What is the reason for this? And why is F102S not studied as an individual mutation?
- In this manuscript, Ibanez de Opakua et al. attempt to identify whether the aggregation-prone regions of Nup98FG are important for the formation of hydrogels. For this, the authors introduce Phe to Ser mutations into GLFG or GLFG-like motifs of the identified aggregating regions and analyze the aggregation behavior of the different mutants by NMR spectroscopy and fluorescence microscopy using thioflavin T (ThT) as a stain.

Unfortunately, the authors do not have an assay for the presence of a hydrogel but rather look at aggregation in a general sense. The authors should rephrase to reflect the experiments undertaken in the article, or include a test for actual hydrogel formation.

- In the discussion, the authors suggest that ‘the tight zipper-like arrangement of the FG-repeats in the [...] fibrils [...] may make it more difficult for nuclear transport receptors to open the FG-meshwork in

order to transport cargo'. The authors, thus, propose distinct regions with different kinds of self-organization that generate specific transport zones. This is an interesting thought, focused on the assumption that amyloid-like interactions persist in the nuclear pore complex. Along with this, the authors should discuss other observations from the recent literature, such as the possibility of a liquid-disordered state of the FG-NUPs inside the transport channel of the nuclear pore complex, the impact of post-translational modifications or other sequence requirements.

- Related to the previous point, the last sentence in the abstract 'ultimately influencing transport processes through the nuclear pore' should be replaced by 'potentially/possibly influencing transport processes...' or another statement reflecting the experiments performed.

Minor comments:

- In the introduction, the authors use 'low complexity domains' as a synonym for 'intrinsically disordered regions'. Intrinsically disordered regions often contain low complexity domains, but they are not synonyms. This should be corrected.
- Reference 27 does not use solid-state NMR to investigate FG gels. Did the authors mean to cite Ader et al., 2010 (doi: 10.1073/pnas.0910163107)?
- Fluorescence images are not usually called 'micrographs' (e.g. Figure 3), this should be corrected.

We would like to express our gratitude to the three referees for their insightful evaluation of our manuscript and for providing us with many helpful comments and suggestions to improve it. We carefully addressed the referees' comments and have made the necessary modifications. We believe the suggested improvements have significantly strengthened the overall quality of the work.

Reviewer #1:

Summary Paragraph: Intrinsically disordered regions (IDRs) are commonly found in proteins that phase separate and are frequently associated with diseases like cancer and neurodegeneration. Because disordered regions lack a predictable three-dimensional structure their precise function in phase separation is difficult to determine. The intrinsically disordered FG-repeat domain of FG-NUP proteins is thought to contribute to the size exclusion barrier formed by the phase separated nuclear pore compartment. The FG repeat of NUP98 also forms fusion oncoproteins whose role in cancers is thought to be driven by its propensity to phase separate. The current study focuses on the role of specific FG-residues in the aggregation of NUP98. Using a combination of site-directed mutagenesis and NMR the authors measure the contribution of distinct residues to the self-association of the protein. Cryo-EM of two different NUP98 peptide fibrils suggests a potential structural basis of nuclear pore transport.

Assessment: This study makes use of state-of-the-art structural biology techniques and applies them to a topic of high interest- how IDRs contribute to phase separation and aggregation. The experiments are of high quality, but some additional quantification is needed. Especially considering NUP98 fusion oncoproteins, those in the fields of cancer, phase separation, and general cell biology will be interested in the current study. With this broad audience in mind, parts of the manuscript should be reorganized for clarity and more experimental detail is needed in the text. I make the following specific suggestions for improvement:

Reply: We thank the reviewer for the positive assessment of our work and for the helpful suggestions for improvements.

1. The rationale for analyzing the F residues chosen is unclear when the solvation energy of other F residues within Nup98FG(85-124) (Figure 1E) and Nup98FG(298-327) (Figure 5A) was approximately equal to or greater than that of F102 and F319.

Reply: We thank the reviewer for this comment. We selected the LF motif because it often shows a large solvation energy. We also agree that there can be some individual residues with larger solvation energies, but these residues are not part of a motif that can easily be compared across the protein sequence. FG motifs are also frequent in the sequence, but we selected LF motifs because they have on average larger solvation energy and also a higher tendency to form beta-structure.

2. Why was the F102S mutation not examined independent of the F319S mutation? This omission makes it difficult to compare the two aggregation-prone regions of Nup98FG.

Reply: We thank the reviewer for this suggestion. We expressed and purified the F102S mutant, and ran all the experiments with that mutant in order to make the comparison easier. The results from these experiments have been incorporated into the various figures and are discussed in the text. Overall, the results from the F102S mutant are consistent with those of other Nup98FG mutant proteins.

3. Quantification of this experiment should be included. Is ThT fluorescence indicative of phase separation or aggregation or both in this context?

Reply: ThT fluorescence is indicative of the presence of beta-amyloid-like structures, which can be correlated either with aggregation or phase separation depending on how one interprets the hydrogel-like particle formation. We propose this figure as qualitative analysis to support the more precise DLS and NMR data.

4. The introduction was very clear and well-written. Suggestions to help with clarity of the results: consider re-organizing the paper to present the cryo-EM structure of the two peptide fibrils at the

beginning. Propose the important residues based on the structure/solvation energy (Figure 1, 4, 5), and then introduce the mutations and show their effect on aggregation (Figures 2 and 3). The results section should also include more experimental detail and explanation to cater to a wider, non-structural biology audience. For example, include one sentence to highlight the utility of each method for the study of IDRs (especially methods used in Figure 2 and the description of the cryo-EM in Figure 4) prior to stating the result. A final schematic that models how the distinct properties of Nup98(85-124) and Nup98(298-327) might contribute to the nuclear pore meshwork/aggregation would be helpful.

Reply: We thank the reviewer for this suggestion. When initialing designing the layout of the manuscript we had both types of organization in mind, i.e. the one proposed by the reviewer and the one ultimately chosen. In the end we settled for the later, because we felt that it best links the previously determined cryoEM structure of the Nup98(85-124) fibrils with the mutational analysis and the newly determined cryoEM structure of Nup98(298-327). We would therefore suggest to stick to the current layout.

Thanks also for suggesting additional sentences to introduce the individual methods. We added them as suggested.

We also very much like the suggestion to add a schematic model at the end of our manuscript. However, previous experience with other manuscripts indicated that some reviewers do not like schematics that may imply to far-reaching hypotheses. For the current manuscript, reviewer #2 already feels that “*The manuscript contains many conclusions that are speculative and are not supported by experimental evidence*”. We therefore believe that it is best to not include additional schematics currently.

Minor:

Figure

1:

1. Given that other types of IDRs contain unique motifs (Ex:RGG repeats), directly comparing the contribution of the FG vs GLFG motif within Nup98FG(298-327)) in the aggregation analysis would be of interest.

Reply: We thank the reviewer for this suggestion. XLFX motifs in general have larger solvation energies (because there are two residues, L and F, with large values) and higher beta-structure content than FG motifs, as shown in our previous paper (Ibáñez de Opakua et al, Nat. Chem. 2022). In the context of the current manuscript focusing on self-association, we believe it is therefore reasonable to keep the focus on the XLFX motifs. Please also note that we already prepared several mutants of Nup98FG and analyzed their self-association propensities with a panel of biophysical methods.

2. The cryo-EM structure in D is very small and is not discussed in detail like the one in Figure 4E.

Reply: Thanks for the suggestion. We added an additional sentence to the revised manuscript that summarizes the key properties of this cryoEM structure. For further details please have a look at our previous publication: Ibáñez de Opakua et al, Nat. Chem. 2022.

Figure

2:

1. Consider including a graph that combines all results at a given pH to make it easier for the reader to directly compare the aggregation of WT and mutants.

Reply: Thanks for the suggestion. We added a figure for the relevant pH 7 (new Fig. 2b).

2. In b there are clearly differences in ellipticity between WT and mutants, but the significance of these differences is unclear.

Reply: In the revised version of the manuscript, we state on page 7: “*Nup98^{FG} is soluble and predominantly disordered at pH 3 with some residual β -structure⁴¹. When compared to the Nup98^{FG} wild-type protein, the amount of β -structure estimated from circular dichroism spectra decreases by ~15% and ~19% for the single (F102S and F319S, respectively) and by 30% for the double (F102S/F319S) mutant Nup98^{FG}, but only by ~7% for the control mutant F228S (Fig. 2c and S1).*”

Figure

3:

1. Do the Nup98FG mutants aggregate less or just on a much slower time scale?

Reply: Thanks for the comment. Currently, we can only say that it aggregates slower. Whether the final level of aggregation is also different would require long-term incubation.

Figure

5:

1. It is unclear why the solvation energy of Nup98FG(298-327) isn't presented until Figure 5 when I believe it's used as rationale for choosing the F319S mutation. Consider adding it to Figure 1 along with the Nup98FG(85-124) data and using the last figure for a model schematic.

Reply: Please see our reply above regarding the organization of the manuscript and the “danger” of adding a schematic.

Reviewer #2:

In this study, Ibáñez de Opakua et al. use nuclear magnetic resonance (NMR) and cryo-electron microscopy (cryoEM) based techniques to probe the structure and biophysical properties of a phenylalanine-glycine (FG) repeat region of the nucleoporin Nup98. The authors determined a 2.67Å cryoEM structure of an amyloid fibril obtained from a Nup98 fragment (residues 298-327) and analyzed the effect of mutations on Nup98's propensity to aggregate by collecting 1H-NMR spectra at different pH values as a proxy for aggregation, circular dichroism, dynamic light scattering, and by staining with the amyloid-binding dye thioflavin-T. The main function of FG repeats in the nuclear pore complex (NPC) is to establish a permeability barrier for the selective transport of macromolecules between the nucleus and the cytoplasm. Intrinsically disordered FG repeat polypeptides line the NPC's transport channel and are thought to form a mesh-like phase that blocks or hinders the free diffusion of proteins and ribonucleic acids in a size-selective manner. In-vitro reconstituted FG repeats have been shown to form liquid-liquid phase separated condensates (Celetti et al., JCB, 2020), which can mature into hydrogels (for example, Frey and Görlich, Cell, 2009 or Schmidt and Görlich, eLife, 2015), and amyloids (Halfmann et al. Prion, 2012; and Alberti et al. Cell, 2009). NMR techniques have been previously used to characterize cohesive FG interactions (Najbauer et al., Nat. Commun., 2022; Labokha et al., EMBO J., 2013; and Ader et al., PNAS, 2010). The current study is a direct extension of the authors' previous work. Previously, Ibáñez de Opakua et al. generated 18 overlapping fragments of a FG repeat region of Nup98 (residues 1-384) which they analyzed by NMR over time (Nature Chemistry, 2022). The rate of NMR signal loss allowed them to score different regions of Nup98 for their tendency to aggregate, and they came up with two “aggregation prone” regions: residues 85-124 and 298-327. In this 2022 study, the authors reported a biophysical analysis of the Nup98 85-124 fragment using NMR, circular dichroism, and physical calculations, and a high-resolution structure of the amyloid fiber obtained by cryoEM. Now, they repeated a similar analysis of the Nup98 298-327 fragment.

Reply: We thank the reviewer for the careful assessment of our work and for the helpful suggestions for improvements.

The manuscript contains many conclusions that are speculative and are not supported by experimental evidence. They should be thus removed. Some examples include: Pages 7-10 (Results and Discussion): The interpretation of the impact of mutations introduced by the authors on Nup98 phase separation. The authors generated the following point mutations: F319S in the “second” aggregation-prone region, F102S/F319S that combines F319S with the F102S mutation in the “first” aggregation-prone region, and F228S in between the two aggregation-prone regions. They found that the Nup98 1-384 fragment is less prone to aggregation if harboring the F102S/F319S mutation. To substantiate the claims that “The combined data demonstrate that destruction of different GLFG motifs by F-to-S mutation decreases the aggregation propensity of Nup98FG. However, a particularly strong effect is exerted by the F319S mutation...” and that “The data show that individual FG-repeats have variable importance for the self-association capabilities of Nup98” the authors would need to analyze the F102S mutant in side-by-side experiments, as well as combinations of F319S with mutations they expect to have a milder effect, such as F228S/F319S. The claims are speculative given the current design of experiments.

Reply: We thank the reviewer for the suggestion to study the F102S mutant. We expressed and purified the F102S mutant, and ran all the experiments with that mutant. The results from these experiments have been incorporated into the figures and are discussed in the text. Overall, the results from the F102S mutant are consistent with those of other Nup98^{FG} mutant proteins. Given these new results and the fact that we already prepared several mutants and analyzed them by a battery of biophysical experiments, we hope that preparation of the F228S/F319S mutant is not essential.

Additionally, we deleted the sentence “The data show that individual FG-repeats have variable importance for the self-association capabilities of Nup98 ...”

Pages 11-12 (Discussion): The authors speculate that the two aggregation-prone regions that they have identified (residues 85-124 and 298-327) assume distinct roles in hydrogel formation and interactions with transport factors based on the packing of hydrophobic residues in the amyloid fibril cryoEM structures they determined. These claims are not substantiated by any experimental evidence, such as systematic mutagenesis of these regions followed by assays for hydrogel formation and transport factor binding. More fundamentally, there is no guarantee that the structures determined by the authors inform us on the interactions that give rise to Nup98 hydrogels in the cell. Work by Scheres et al. and by the authors themselves on the Nup98 85-124 fragment has shown that the same polypeptides aggregate into a variety of amyloid fiber structures depending on the aggregation conditions.

Reply: We thank the reviewer for these comments. We fully agree with the reviewer that the suggested roles of the two aggregation-prone regions is only a hypothesis which we raised in the discussion section. To clarify the speculative nature of this hypothesis and stress the need for further experiments, we added clarifying statements to the discussion in the revised version of the manuscript:

“The large structural differences, which we observe in the amyloid-conformations of the two different FG-rich regions of Nup98, have a small effect on the hydrogel-like particle formation behavior of Nup98^{FG}. The F102S mutant presents a smaller decrease in β -structure than F319S (also detectable by the appearance of larger particles by DLS) and keeps slightly less monomeric protein in our NMR experiments. This change goes in the direction of the fact that the Nup98(85-124) fibrils have less packed side-chain conformations and display lower stability per residue (-0.51 kcal/mol) when compared to Nup98^{FG}(298-327) fibrils (-0.65 kcal/mol).” And

“However, further in vitro and in vivo experiments will be required to confirm these hypotheses.”

Page 10 (Discussion): There is no clear link between the presence of Nup98 in tau aggregates associated with Alzheimer’s disease or the oncogenic Nup98 fusions and the aggregates or amyloid fibrils studied by the authors, so it is not clear how this study advances the understanding of Nup98’s role in these diseases.

Reply: We believe that this comment may arise from our statement at the end of paragraph 1 of the discussion section “Detailed molecular insights into the self-association of the FG-repeat domain of Nup98 are thus critical to better understand its diverse physiological and disease-associated activities.” In ref #46 it was shown that the FG-repeat domain of Nup98 facilitate the aggregation of the microtubule-associated protein tau, resulting in the formation of tau-containing Nup98 aggregates in neuron. Currently, we do not know in which form the Nup98 FG-repeats are present in these aggregates and whether these are co-assemblies with tau or each of the two proteins segregate into “homotypic” fibrils. To gain insight into these questions many more experiments are required. However, our structure shows interactions that Nup98 can make to form fibrils and these interactions may be relevant also in the context of Nup98 in tau aggregates associated with AD. Please note that we did not do any such specific speculations in our manuscript.

A significant issue with the manuscript is that the referencing of previous work is incomplete and overall arbitrary. Some examples include: Page 2 (Introduction): In general, because the authors studied the human Nup98, all citations discussed in the introduction to the NPC should reference the human NPC instead of the *S. cerevisiae* NPC. For example, Kim et al. (Nature, 2018) is not appropriate for the claim “Each NPC contains multiple copies of over 10 FG-NUPs...”.

Page 2 (Introduction): Lin et al. (Science, 2016) should be cited in relation to “The nucleoporins situated in the central channel of NPCs are known as phenylalanine/glycine (FG)-NUPs due to their consecutive repeat of the FG motif.”

Page 2 (Introduction): The claim that “This hydrogel allows low molecular weight proteins and nuclear transport receptors to pass while impeding the passage of proteins weighing above 30 kDa.” is too simplistic for a manuscript about FG properties. The authors should consult Ribbeck et al. (EMBO J, 2001)

Page 3 (Introduction): The reference #27 (Milles et al., EMBO Rep., 2013) is not applicable to the claim “Solid-state NMR analysis of nucleoporin FG gels further...”. Is this a typo?

Page 3 (Introduction): The references associated with the claim that “structural analyses of NPCs using high-speed atomic force microscopy and cryo-electron tomography demonstrated that FG-NUPs extend filamentous protrusions into the central channel” do not actually “demonstrate” that what had been observed is an FG nucleoporin per se.

Reply: We thank the reviewer for the suggestion to introduce/correct references. We introduced these references, removed the sentence about the passage of small/large molecules, and changed “demonstrated” to “suggested” in the revised version of the manuscript.

The authors need to discuss, in the introduction and discussion sections, the previously determined amyloid structure of a Nup98 fragment (residues 116-123) by Hughes et al. (2018, Science) and how the interactions captured by this fragment relates to the structures obtained from the Nup98 85-124 and 298-327 fragments.

Reply: We thank the reviewer pointing out the publication by Hughes et al.. We added corresponding statements to the introduction and discussion sections.

Lastly, the physiological relevance of the Nup98 FG repeat structures determined by Ibáñez de Opakua et al. in the current and their previous manuscript (Nature Chemistry, 2022) needs to be established. Under what conditions do these particular amyloid fibers occur in the human body? Transport factors and chaperones have been shown to prevent the formation of FG repeat aggregates (Springhower, Rosen, and Chook, 2020, Curr. Opin. Cell Biol.; Prophet et al., 2022, Nat. Cell Biol.; and Kuiper et al., 2022, Nat. Cell Biol.). The structural variability of different Nup98 FG-repeat region fragments, as well as variability of structures that arise from the same 85-124 fragment, raises the question of what the actual structure of the intact FG repeat region is and what structures can occur in vivo. The lack of physiological validation substantially diminishes the excitement stemming from these structures, as it is unclear what can be learned about NPC (dys)function.

Reply: We agree with the reviewer that further studies are required to understand the structure of Nup98FG repeats (and other Nup FG repeats) in the context of the intact nuclear pore. Please note however that Nup98 FG repeats also play a role in cancer-related condensates and form aggregates together with tau in neurons. Thus, while chaperones and transport factors may prevent formation of FG repeat aggregates inside the nuclear pore, the formation of amyloid-like Nup98 FG aggregates related to onco-condensates and in the context of neurodegeneration is possible. Because of the eminent lack of structural knowledge about Nup98 FG aggregates, our work provides first important insights into structural arrangements and the underlying molecular interactions of Nup98 FG aggregates.

The currently presented data, which includes a structure of an amyloid fiber obtained from a Nup98 FG-repeat fragment (residues 298-327) and a mutational analysis that lacks rigor, is a modest advance with little conceptual novelty over the previous data, which included a mapping of aggregation propensity for the entire FG repeat region as well as structures and a mutational analysis of a different Nup98 FG-repeat fragment (residues 85-124). Many conclusions represent speculations that are not supported by experimental evidence. Unfortunately, the results only provide an incremental advance from previously published work, making this paper more appropriate for a more specialized journal.

Reply: We thank the reviewer for allowing us to add further mutational analysis to the manuscript. As described above, we expressed and purified the F102S mutant, and ran all the experiments with this mutant. The results from these experiments have been incorporated into the figures and are discussed in the text.

Overall, the results from the F102S mutant are consistent with those of the other Nup98FG mutant proteins. Additionally, we removed some of the speculations from the revised version of the manuscript.

Please note that the cryoEM structure of Nup98FG(298-327) is only the second high-resolution structure of an amyloid fibril formed by a longer region of Nup98FG than a very short peptide (i.e. Nup98 116-123). It thus provides insights into the molecular interactions that drive self-association and stabilize amyloid-like fibril structures of Nup98FG. The cryoEM structure of Nup98FG(298-327) is also distinct from that of Nup98FG(85-124) providing insight into the spectrum of molecular arrangements FG-repeats can adopt inside aggregate structures.

Reviewer #3:

Ibanez de Opakua et al. present a manuscript in which they analyze the aggregation of the FG-nucleoporin Nup98. Based on aggregation assays using small Nup98 peptides, the authors calculate a per-residue aggregation propensity and thereby identify two regions within the protein that they predict to be particularly prone to aggregate. Through the insertion of specific Phe to Ser mutations into the aggregation prone regions (and the region in between), the authors aim to address whether GLFG motifs, important for hydrogel formation and nucleocytoplasmic transport, are also important for self association. Several aggregation assays and assays addressing secondary structure content (1D ¹H NMR, CD spectroscopy, DLS, ThT staining and imaging of aggregates) are presented and confirm that Phe to Ser mutations decrease the aggregation propensity. Nup98 peptides comprising the two aggregation prone regions, respectively, were shown to form fibers and the structure of those was analyzed using electron microscopy. A detailed structural analysis of the fibers formed by the second aggregation prone peptide is presented and reveals the participation of the GLFG motif. The authors discuss the structural differences between the fibers formed by the two aggregating peptides and their potential impact for accessibility of FG-repeats to nuclear transport receptors and in the context of oncogenic fusion proteins. The authors present a solid piece of work studying the aggregation of Nup98 by different methods and providing molecular insights into its aggregation behavior, which is of potential relevance to nucleocytoplasmic transport and oncogenic fusion condensates.

Reply: We thank the reviewer for the positive assessment of our work and for the helpful suggestions for improvements.

However, a few points need to be addressed:

Based on an aggregation assay developed in Ibanez de Opakua et al. (2022), the authors calculate a per residue aggregation propensity. As the peptides used for this assay seem to be of different length and only partially overlapping, it is not clear how this has been done. Can the authors report how the per residue aggregation propensity was calculated?

Reply: We thank the reviewer for encouraging us to clarify the approach. In the methods section of the revised version of the manuscript, we now added: *“For the estimation of aggregation propensities, we obtained the percentage of aggregation after one day for each peptide in the NMR measurement conditions. The presented values per residue, when overlapping, are the average values of the peptides with that residue.”*

Two aggregation prone regions are identified, of which one is very similar to the one already studied in Ibanez de Opakua et al. (2022). The authors identify the structure of the amyloid fibers formed by that peptide, but do not report on any of their features or further analysis. If the authors decide to report on the newly determined structure of Nup98 85-124, they should dedicate more than one sentence to its analysis.

Reply: Please note that the structure presented in Fig. 1 is identical to that determined in Ibanez de Opakua et al. (2022). In the revised version of the manuscript, we now added the statement: *“This structure contains a large cavity formed by polar residues and most of the GLFG-like motifs fold into a β -turn/ β -arch structure with phenylalanine and leucine pointing to the same side.”*

This is particularly true, when the structural implications of the mutations are discussed, which is focused on the second peptide and F319S. No particular attention is given to F102S. What is the reason for this? And why is F102S not studied as an individual mutation?

Reply: We thank the reviewer for these suggestions. For the revision of our manuscript, we expressed and purified the F102S mutant, and ran all the experiments with that mutant. The results from these experiments have been incorporated into the various figures and are discussed in the text. Overall, the results from the F102S mutant are consistent with those of other Nup98^{FG} mutant proteins.

In this manuscript, Ibanez de Opakua et al. attempt to identify whether the aggregation-prone regions of Nup98^{FG} are important for the formation of hydrogels. For this, the authors introduce Phe to Ser mutations into GLFG or GLFG-like motifs of the identified aggregating regions and analyze the aggregation behavior of the different mutants by NMR spectroscopy and fluorescence microscopy using ThT as a stain. Unfortunately, the authors do not have an assay for the presence of a hydrogel but rather look at aggregation in a general sense. The authors should rephrase to reflect the experiments undertaken in the article, or include a test for actual hydrogel formation.

Reply: We thank the reviewer for this comment. In Fig. 3 we used microscopy to observe Nup98^{FG} particles which may be described as hydrogel-like particles. Additionally, we detected larger particles (~130 and 240 nm of hydrodynamic radius) by DLS. These particles were previously suggested to have hydrogel-like properties (ref #22, 25). We would therefore use the term “hydrogel-like particles” throughout the revised version of the manuscript.

In the discussion, the authors suggest that ‘the tight zipper-like arrangement of the FG-repeats in the [...] fibrils [...] may make it more difficult for nuclear transport receptors to open the FG-meshwork in order to transport cargo’. The authors, thus, propose distinct regions with different kinds of self-organization that generate specific transport zones. This is an interesting thought, focused on the assumption that amyloid-like interactions persist in the nuclear pore complex. Along with this, the authors should discuss other observations from the recent literature, such as the possibility of a liquid-disordered state of the FG-NUPs inside the transport channel of the nuclear pore complex, the impact of post-translational modifications or other sequence requirements.

Reply: We thank the reviewer for this suggestion. In the revised version of the discussion, we added the possibility of liquid-disordered states of FG-NUPs inside the transport channel, as well as the potential regulation of FG-NUP properties by post-translational modifications such as O-GlcNAcylation.

Related to the previous point, the last sentence in the abstract ‘ultimately influencing transport processes through the nuclear pore’ should be replaced by ‘potentially/possibly influencing transport processes...’ or another statement reflecting the experiments performed.

Reply: Thanks, we changed it as suggested.

Minor comments:

In the introduction, the authors use ‘low complexity domains’ as a synonym for ‘intrinsically disordered regions’. Intrinsically disordered regions often contain low complexity domains, but they are not synonyms. This should be corrected.

Reference 27 does not use solid-state NMR to investigate FG gels. Did the authors mean to cite Ader et al., 2010 (doi: 10.1073/pnas.0910163107)?

Fluorescence images are not usually called ‘micrographs’ (e.g. Figure 3), this should be corrected.

Reply: Thanks. We corrected the issues as suggested.

REVIEWER COMMENTS

Reviewer #1 (Remarks to the Author):

The authors have adequately addressed all concerns except one: as requested in the first review, I would still ask for quantification of the ThT staining experiment to better support the statement at the bottom of page 7, "Nup98FG proteins that carry mutations in the two aggregation-prone regions form a smaller number of less clustered particles." Once this is added the paper is suitable for publication.

Reviewer #2 (Remarks to the Author):

The authors have done a commendable job in addressing technical aspects of the review. In particular, the authors have addressed the missing characterization of the F102S mutation, producing results that would be expected based on the analogous F319S mutation and the combination F102S/F319S mutation. The authors have addressed all concerns about missed references and properly qualified some of their more speculative statements as such.

However, it should also be noted that these additional results have not yielded further conceptual insight. As the authors have pointed out in the rebuttal, the structure is the second high-resolution structure of an amyloid fibril formed by a longer region of Nup98FG. Whereas it might provide insight into the molecular interactions that drive Nup98FG self-association and stabilize amyloid-like fibrils, there is also no evidence that the determined structure and in vitro biophysical characterization of these Nup98FG fragments have any relevance to Nup98's physiological functions or its contributions to the pathology of tau fibers or Nup98 oncogenic fusions.

Reviewer #3 (Remarks to the Author):

In their revised version, Ibáñez de Opakua et al. addressed all my comments and, with the study of mutation F203S also significantly strengthened their manuscript. Only a few small points remain to be addressed:

- Concerning the biological interpretation of their results, which are without doubt of potentially high significance for nucleocytoplasmic transport and disease, I would encourage the authors to remain a little more hypothetical, particularly in the abstract ('potentially influencing transport processes' rather than 'ultimately' for example).

- Even though the authors have changed 'hydrogel' into 'hydrogel-like particles', I still think, the

manuscript should talk about aggregation in a more general sense, since assays with respect to hydrogel formation have not been undertaken. This is different for the introduction, in which the authors cite articles, which have shown the presence of actual hydrogels, which accumulate transport receptors/transport complexes. Aggregation properties are certainly relevant to hydrogel formation and phase separation, but those are not studied in the presented manuscript.

- In particular in the beginning of the results section the authors should make more clear what has been done already in their previous paper compared to this manuscript (e.g. the design of the 18 peptides).

- It is still not fully clear, how residue wise aggregation propensity was calculated, since the peptides differ by more than one residue. Were two-dimensional NMR spectra recorded and residue wise aggregation assessed? If so, this should be clearly stated.

- How was the 'prion-like domain (PrD-like) propensity' calculated? A reference seems to be missing.

We would like to express our gratitude to the three referees for their insightful evaluation of our manuscript and for providing us with many helpful comments and suggestions to improve it. We carefully addressed the referees' comments and have made the necessary modifications. We believe the suggested improvements have significantly strengthened the overall quality of the work.

Reviewer #1:

The authors have adequately addressed all concerns except one: as requested in the first review, I would still ask for quantification of the ThT staining experiment to better support the statement at the bottom of page 7, "Nup98FG proteins that carry mutations in the two aggregation-prone regions form a smaller number of less clustered particles." Once this is added it is suitable for publication.

Reply: We thank the reviewer for the positive assessment of our work. Regarding the ThT staining experiment, we used it as a qualitative experiment to show the aspect of the formed particles for each mutant. For a quantitative analysis we used the NMR and DLS data. In the newly revised version of the manuscript, we removed the sentence cited by the reviewer, which suggesting quantitative conclusions from a qualitative experiment and, as suggested by the reviewer, that is not correct. Thanks for this correction.

Reviewer #2:

The authors have done a commendable job in addressing technical aspects of the review. In particular, the authors have addressed the missing characterization of the F102S mutation, producing results that would be expected based on the analogous F319S mutation and the combination F102S/F319S mutation. The authors have addressed all concerns about missed references and properly qualified some of their more speculative statements as such.

Reply: We thank the reviewer for the positive assessment of our work.

However, it should also be noted that these additional results have not yielded further conceptual insight. As the authors have pointed out in the rebuttal, the structure is the second high-resolution structure of an amyloid fibril formed by a longer region of Nup98FG. Whereas it might provide insight into the molecular interactions that drive Nup98FG self-association and stabilize amyloid-like fibrils, there is also no evidence that the determined structure and in vitro biophysical characterization of these Nup98FG fragments have any relevance to Nup98's physiological functions or its contributions to the pathology of tau fibers or Nup98 oncogenic fusions.

Reply: We believe that the presented work addresses a rather unknown issue, such as the molecular interactions occurring within these disordered structures that form the interior of the nuclear pore and of oncogenic fusion condensates. Consequently, the structure presented provides a new focal point for studying this unfamiliar subject. However, as pointed out by the reviewer, further evidence is needed to demonstrate this relationship. Therefore, we modified the last sentence at the end of the discussion to clarify this aspect:

"However, further in vitro and in vivo experiments will be required to confirm these hypotheses and especially to establish the physiological role of these Nup98 fibrillar structures."

Reviewer #3:

In their revised version, Ibáñez de Opakua et al. addressed all my comments and, with the study of mutation F203S also significantly strengthened their manuscript. Only a few small points remain to be addressed:

Reply: We thank the reviewer for the careful assessment of our work and for the helpful suggestions for improvements.

Concerning the biological interpretation of their results, which are without doubt of potentially high significance for nucleocytoplasmic transport and disease, I would encourage the authors to remain a little more hypothetical, particularly in the abstract ('potentially influencing transport processes' rather than 'ultimately' for example).

Reply: We adapted the text as suggested.

Even though the authors have changed 'hydrogel' into 'hydrogel-like particles', I still think, the manuscript should talk about aggregation in a more general sense, since assays with respect to hydrogel formation have not been undertaken. This is different for the introduction, in which the authors cite articles, which have shown the presence of actual hydrogels, which accumulate transport receptors/transport complexes. Aggregation properties are certainly relevant to hydrogel formation and phase separation, but those are not studied in the presented manuscript.

Reply: We adapted the text as suggested. We updated all references to our particles as hydrogel-like particles to aggregated particles or aggregates.

In particular in the beginning of the results section the authors should make more clear what has been done already in their previous paper compared to this manuscript (e.g. the design of the 18 peptides).

Reply: We used a reference when talking about that experiment to show that the data comes from there. To make it more clear we added "data from our previous publication" together with the reference.

It is still not fully clear, how residue wise aggregation propensity was calculated, since the peptides differ by more than one residue. Were two-dimensional NMR spectra recorded and residue wise aggregation assessed? If so, this should be clearly stated.

Reply: We used just the peptide data. The aggregation propensity of each residue is determined by averaging the aggregation propensities of the peptides containing that residue. We updated the methods section with this more clear explanation.

How was the 'prion-like domain (PrD-like) propensity' calculated? A reference seems to be missing.

Reply: We used the PLAAC server for that calculation. We added the reference in the figure legend.

REVIEWERS' COMMENTS

Reviewer #3 (Remarks to the Author):

The authors have addressed all my remaining comments and provide a very clear piece of work on the aggregation of Nup89.